# Class-Incremental Continual Learning for Multi-View Clustering

## Abstract

Multi-view clustering (MVC) aims to explore common semantics for multi-view data and has become an active research topic. However, existing MVC methods focus on learning from static training data and ignore streaming multi-view data with incremental classes, which is frequent in real-world applications given the continually evolving nature of our world. Meanwhile, the existing continual clustering methods only consider single-view data, which cannot effectively mine the semantics of multi-view data. In this paper, we propose a novel Class-incremental Continual Multi-View Clustering (CCMVC) method to handle class-incremental continual learning for multi-view clustering, where multi-view data with incremental semantic classes come sequentially. Our method conducts two iterative optimization phases, i.e., multi-view cluster search and multi-view cluster consolidation, for sequential multi-view training data. In the test, our CCMVC can perform online multi-view clustering for all emerged classes. Firstly, CCMVC learns the common feature space for multi-view data and searches clusters for the incoming data. Secondly, CCMVC harmonizes and consolidates all learned clusters in a unified MVC model with data replay for all emerged classes. In particular, we propose a cross-view synchronous loss to mitigate the asynchronous convergence problem inherent in multi-view continual learning. Extensive experiments on six public MVC datasets reveal the superiority of CCMVC compared with the state-of-the-art methods.

## 1 Introduction

With the rapid development of Internet and multimedia, multi-view data (e.g., social media posts with texts and images) have been increasingly frequent. Therefore, the analysis of multi-view data has also become an important research topic. Multi-View Clustering (MVC) aims to explore the semantic structure of multi-view data and cluster multi-view data into different classes, where class labels are unavailable during training (Zhou & Shen, 2020; Trosten et al., 2021; Xu et al., 2022). Though researchers have made significant progress in single-view clustering (MacQueen, 1967; Ester et al., 1996; Li et al., 2021), the multi-view nature of MVC makes the problem different. Besides the richer semantics brought by multiple views, the heterogeneity incurred by the inconsistent distributions of multi-view features can impact the effectiveness of clustering. Therefore, the objective of multi-view clustering is to overcome the heterogeneity and discover the common clustering structure by learning from all available views, simultaneously.

Current MVC methods (Xu et al., 2022; Tang & Liu, 2022; Hu et al., 2023) primarily focus on clustering static multi-view data, where the number of semantic classes remains unchangeable, and learning the clusters at once on a fixed dataset. However, new data with new semantic classes can continually emerge in real-world applications. In such a situation, existing MVC models have to learn clusters for all data from scratch and cannot utilize the already learned knowledge, which incurs redundant computation costs and slows down the response to new data. Even worse, in some scenarios, the model is not allowed to access the complete past data due to limitations such as privacy problems. To handle the above problems, in this paper, we study Class-incremental Continual Learning for Multi-View Clustering, which aims at continually learning incremental clusters for streaming multi-view data of incremental classes. Continual learning (CL) (De Lange et al., 2022; Kumar et al., 2021b) is the paradigm where a single model is required to learn a sequence of tasks. Specifically, in the $t$-th task of class-incremental continual learning (Li & Hoiem, 2016; Kirkpatrick

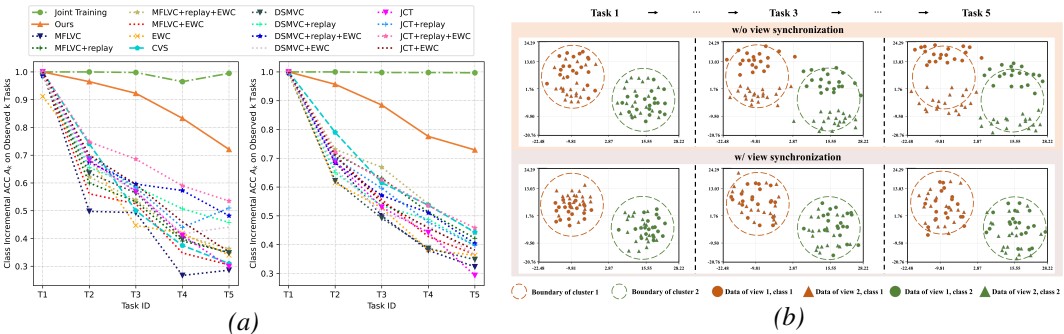

Figure 1: (a) Class incremental accuracy $A_k$ on Fashion (left) and MNIST-USPS (right) datasets. $A_k$ denotes ACC on all past $k$ tasks. (b) Visualization of asynchronous problem across different views in multi-view continual learning on MNIST-USPS dataset.

et al., 2017; Wan et al., 2022a), a CL model trained on the previous $(t-1)$ tasks is required to continually learn clusters for the $t$-th incoming dataset of newly emerging classes. In the test, the CL model is required to perform previous $t$ tasks jointly without task identifiers by using the newly learned knowledge for the $t$-th task and reusing the already learned knowledge for previous $(t-1)$ tasks. Despite the great application value of continual multi-view clustering, it is still an under-explored problem. To the best of our knowledge, this is the first work to study class-incremental continual learning for multi-view clustering.

Applying the existing MVC methods to continual learning scenarios usually suffers from the catastrophic forgetting problem, where the model forgets the already learned knowledge for previous tasks while learning a new clustering task. As shown in Figure 1 (a), we compute the class incremental accuracy $A_k$, which is the accuracy of performing all past $k$ clustering tasks jointly, for compared methods during continual learning. $A_k$ of the state-of-the-art MVC methods MFLVC (Xu et al., 2022), DSMVC (Tang & Liu, 2022), and JCT (Hu et al., 2023) drops significantly while learning more tasks, leaving a large gap between Joint Training which is trained from scratch on joint $k$ datasets. These results reveal that directly applying existing methods to continual MVC can lead to catastrophic forgetting, where the models forget the knowledge for performing previous tasks. Therefore, class-incremental continual learning brings up a new challenge for MVC: **(1) How to continually learn to cluster streaming multi-view data with incremental classes and prevent catastrophic forgetting?**

Furthermore, the core of continual learning is dynamically updating the model to learn new knowledge and prevent forgetting the learned knowledge. Existing work for continual clustering only considers clustering single-view data (Rao et al., 2019; Kumar et al., 2021a; Korycki & Krawczyk, 2021). However, the multi-view nature makes the problem of continual learning for MVC more difficult. In multi-view learning, researchers have found that multi-view features can have asynchronous convergence rates in learning (Wang et al., 2020; Winterbottom et al., 2020). The sequential learning process of continual learning makes the problem more tricky, where imbalanced update rhythm across views can be accumulated and amplified to make multi-view features misaligned. As shown in the upper of Figure 1 (b), without the view synchronization strategy proposed in this paper, the features of different views may be separated during the continual learning process, which degenerates the learned cluster structure. Therefore, we have to address a new challenge for multi-view continual learning: **(2) How to balance the learning rhythm across different views while learning a new task and memorizing already learned tasks?**

To overcome the above challenges, we propose a Class-incremental Continual Multi-view Clustering (CCMVC) method to handle class-incremental continual learning for MVC, which can continually learn incremental clusters for streaming multi-view data and memorize learned knowledge to perform all learned clustering tasks simultaneously. For **Challenge (1)**, we propose a continual component expansion with self-supervised data replay to continually expand the clustering components and prevent catastrophic forgetting. To represent the incremental semantic classes during continual learning, we learn the clustering components for the current task and dynamically expand the component pool for all past tasks. To prevent catastrophic forgetting, we maintain a memory to save a few samples of the past tasks and generate the pseudo labels according to their past clustering

results. Then, we conduct the data replay with classification loss on pseudo labels to consolidate all learned clustering components. For **Challenge (2)**, we propose a cross-view synchronous loss to balance the learning rhythm across different views during continual learning. By using pseudo labels to construct self-supervised contrastive losses across views, we can force minimizing the learning difference between different views based on learned knowledge. Therefore, we can keep the similarity between past data and obtain more synchronous multi-view learning rhythms. Comprehensive experimental results on six MVC datasets demonstrate a significant performance improvement of our proposed model compared with the state-of-the-art methods.

In brief, the contributions of this paper are listed as follows:

- To the best of our knowledge, this work is the first to study class-incremental continual learning for multi-view clustering. Besides empirically showing that existing MVC methods fail to handle class-incremental continual learning, we propose a novel CCMVC method to cluster multi-view data while new semantic classes continually emerge in streaming training data.

- We propose a continual component expansion with self-supervised data replay to continually learn incremental clustering components and prevent catastrophic forgetting. We maintain a data memory and conduct data replay with self-supervised pseudo labels after learning a new task to consolidate all learned clustering components.

- We propose a cross-view synchronous loss to balance the learning rhythm across different views during continual learning. By developing cross-view self-supervised contrastive learning, we minimize the difference between multi-view features of the same cluster to keep the alignment across different views.

- Extensive experiments on six public datasets demonstrate a significant performance improvement of the proposed CCMVC compared with both multi-view clustering baselines and continual learning baselines.

## 2   RELATED WORK

**Multi-view clustering.** Multi-View Clustering (MVC) aims to explore the semantic structure of multi-view data and cluster multi-view data. For instance, Zhang et al. (2021) obtain partition representations of each view through deep matrix decomposition, which are jointly utilized with the optimal partition representation. Subspace clustering methods (Luo et al., 2018; Li et al., 2019) focus on learning a common subspace representation for multiple views. For example, DiMSC (Cao et al., 2015) extends the traditional subspace clustering to MVC by proposing a diversity term to explore the complementarity of multi-view representations. Deep MVC methods (Zhou & Shen, 2020; Trosten et al., 2021) utilize deep learning models to learn effective features for multi-view data. Among deep methods, MFLVC (Xu et al., 2022) is the state-of-the-art method that learns different levels of features and achieves the reconstruction objective and consistency objectives in different feature spaces. Different from the abovementioned methods that focus on clustering static data with fixed semantic classes, we study the class-incremental continual learning for MVC where new data and new classes continually emerge in real-world applications.

**Continual learning.** Continual Learning (CL) studies the problem of learning from streaming data, with the goal of gradually extending acquired knowledge and using it for future learning (De Lange et al., 2021; Madaan et al., 2022). The major challenge in CL is to learn without catastrophic forgetting: performance on a previously learned task should not significantly degrade over time as new tasks are added. For example, Co2L (Cha et al., 2021) proposes a rehearsal-based algorithm that focuses on continually learning and maintaining transferable representations. CL has also been introduced to single-view clustering. For example, Abhishek et al. (Kumar et al., 2021a) propose a Bayesian VAE to learn the deep structure for each task, which can support inter-task transfer through weight overlapping. Recently, Wan et al. (Wan et al., 2022b) study view-incremental continual learning for MVC, where new views continually emerge during training, and propose a late fusion framework. Differently, we study a more practical problem of class-incremental continual learning for MVC, where new data with new semantic classes continually emerge during training. To the best of our knowledge, this is the first work to study class-incremental continual learning for MVC.

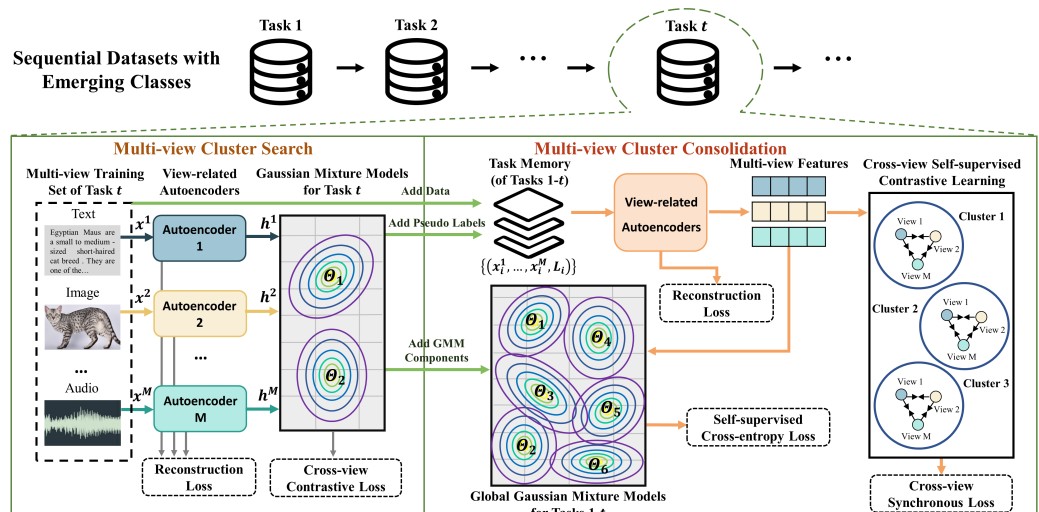

Figure 2: Overview of the proposed CCMVC model: (1) Multi-view training data of task $t$ are projected into a common feature space by view-related autoencoders; (2) Multi-view features are utilized to learn Gaussian mixture models with cross-view contrastive loss for task $t$; (3) A small number of multi-view data with pseudo labels of task $t$ are added into the task memory; (4) Gaussian components for task $t$ are added into the global Gaussian mixture models to perform clustering for tasks 1-$t$ jointly; (5) Data in the task memory are replayed to compute reconstruction loss, self-supervised cross-entropy loss, and cross-view synchronous loss to consolidate all learned clusters.

## 3    NOTATION AND PROBLEM DEFINITION

In this paper, we explore **class-incremental continual learning** (Li & Hoiem, 2016; Kirkpatrick et al., 2017; Wan et al., 2022a) for multi-view clustering. For a growing unlabeled multi-view data sequence, the set of the sequentially collected training data is $\mathfrak{T} = \{\mathcal{T}_1, \mathcal{T}_2, ..., \mathcal{T}_T\}$. The training set of the $t$-th task is defined as $\mathcal{T}_t = \{\boldsymbol{X}_t^m \in \mathbb{R}^{N \times D_m}\}_{m=1}^{M}$ which contains $N$ samples with $M$ views. We denote $\mathcal{C}_t$ as the set of the semantic classes in $\mathcal{T}_t$, which satisfies $\mathcal{C}_i \cap \mathcal{C}_j = \varnothing, \forall i \neq j$. The model is required to learn from the sequential training data $\mathcal{T}_1, \mathcal{T}_2, ..., \mathcal{T}_t$ and cluster multi-view test data $\widetilde{\mathcal{T}}_t$ belonging to the classes $\bigcup_{i=1}^{t} \mathcal{C}_i$ of all past tasks after learning the $t$-th task. Note that in class-incremental continual learning, task identifiers (or task labels) are unavailable in the test. Therefore, we have to learn a unified model for all past classes. While learning the $t$-th task, the model is only allowed to access the $t$-th training set $\mathcal{T}_t$ and memory, where memory preserves a small number of past samples in $\bigcup_{i=1}^{t-1} \mathcal{T}_i$. Therefore, a key challenge of continual learning is to prevent forgetting clusters learned from the previous $(t-1)$ tasks while learning the $t$-th task.

## 4    METHODOLOGY

The overall architecture of our approach is demonstrated in Figure 2, which consists of **Multi-view Cluster Search (MCS)** and **Multi-view Cluster Consolidation (MCC)** during learning the $t$-th task. **MCS** updates the multi-view autoencoders for extracting multi-view common features and learns the cluster components for the current training set $\mathcal{T}_t$. **MCC** conducts continual component expansion with self-supervised data replay to continually expand the clustering components and prevent catastrophic forgetting. Moreover, we propose a cross-view synchronous loss in MCC to balance the learning rhythm across different views during continual learning.

### 4.1    MULTI-VIEW CLUSTER SEARCH (MCS)

While learning the $t$-th task, the multi-view cluster search aims at updating the multi-view autoencoders for multi-view feature extraction and learning clustering components $\Theta_t = \{(\boldsymbol{\mu}_i, diag(\boldsymbol{\sigma}_i^2))\}_{i=1}^{|\mathcal{C}_t|}$ of Gaussian mixture models (GMM) for the $t$-th training set $\mathcal{T}_t$.

### 4.1.1 VIEW-RELATED AUTOENCODERS

Following previous work (Xu et al., 2022), we adopt autoencoders (Baldi, 2012) to project multi-view data into a common space. Specifically, we construct an encoder $E^m(\cdot; \zeta^m)$ and a decoder $D^m(\cdot; \phi^m)$ for the $m$-th view, which are both implemented by multi-layer perceptron (Kim & Adali, 2002). For $\forall \boldsymbol{x^m} \in \boldsymbol{X}_t^m$ of the $m$-th view where $\boldsymbol{X}_t^m \in \mathcal{T}_t$, it is projected into a $D$-dimensional feature space and then reconstructed as follows:

$$
\begin{aligned}
\boldsymbol{h}^m &= E^m(\boldsymbol{x}^m; \zeta^m) \in \mathbb{R}^D, m = 1, ..., M, \\
\hat{\boldsymbol{x}}^m &= D^m(\boldsymbol{h}^m; \phi^m) \in \mathbb{R}^{D_m}, m = 1, ..., M,
\end{aligned}
\tag{1}
$$

where $\zeta^m$ and $\phi^m$ are trainable parameters for the $m$-th autoencoder and we denote $\boldsymbol{\zeta} = \{\zeta^m\}_{m=1}^M$, $\boldsymbol{\phi} = \{\phi^m\}_{m=1}^M$. In a continual learning manner, $\boldsymbol{\zeta}$ and $\boldsymbol{\phi}$ are shared across all tasks to form a union model for performing all learned tasks simultaneously. To learn semantics in the $t$-th task and avoid model collapse, we adopted the unsupervised reconstruction loss to update multi-view autoencoders as follows:

$$
\mathcal{L}_{rec} = \frac{1}{M} \sum_{m=1}^M \mathbb{E}_{\boldsymbol{x}^m \in \boldsymbol{X}_t^m} \{ \| \boldsymbol{x}^m - \hat{\boldsymbol{x}}^m \|_2^2 \}.
\tag{2}
$$

Based on the learned features $\boldsymbol{H}^m = \{\boldsymbol{h}^m | \boldsymbol{x}^m \in \boldsymbol{X}_t^m\}, m = 1, ..., M$, we aim to conduct multi-view clustering for the $t$-th task and mine the common clustering components across all views.

### 4.1.2 CROSS-VIEW CONTRASTIVE LEARNING FOR GAUSSIAN MIXTURE MODELS

We cluster multi-view features by Gaussian mixture models (GMM) (Pernkopf & Bouchaffra, 2005) with components $\Theta_t = \{(\boldsymbol{\mu}_i, diag(\boldsymbol{\sigma}_i^2))\}_{i=1}^{|\mathcal{C}_t|}$. For $\forall \boldsymbol{h}^m \in \boldsymbol{H}^m$ where $m = 1, ..., M$, the generative probability of GMM can be computed as follows:

$$
y_{(i)}^m = \mathcal{N}(\boldsymbol{h}^m | \boldsymbol{\mu}_i, diag(\boldsymbol{\sigma}_i^2)), i = 1, ..., |\mathcal{C}_t|,
\tag{3}
$$

where $y_{(i)}^m$ denotes the probability of the $i$-th Gaussian component generating $\boldsymbol{h}^m$, and we denote $\boldsymbol{y}^m = GMM(\boldsymbol{h}^m; \Theta_t) = [y_{(1)}^m, ..., y_{(|\mathcal{C}_t|)}^m] \in [0, 1]^{|\mathcal{C}_t|}$. For a specific sample, features of different views are supposed to belong to the same semantic class. Moreover, inspired by the previous work about contrastive clustering (Li et al., 2021; Zhong et al., 2021), we further design cross-view dual-anchor contrastive learning on $\boldsymbol{Y}^m = \{\boldsymbol{y}^m | \boldsymbol{x}^m \in \boldsymbol{X}_t^m\}(m = 1, ..., M)$ to align the generative probability across multiple views. Since labels of the same sample should be consistent across views, the cross-view contrastive loss between $\boldsymbol{Y}^m$ of the $m$-th view and $\boldsymbol{Y}^l$ of the $l$-th view with sample-anchor is formulated as follows:

$$
\ell_{ml}^{sa} = \mathbb{E}_{\boldsymbol{x}^m \in \boldsymbol{X}_t^m} \Big\{ -log \frac{exp(cos(\boldsymbol{y}^m, \boldsymbol{y}^l)/\tau_s)}{\sum_{\boldsymbol{x}_*^l \in \boldsymbol{X}_t^l} exp(cos(\boldsymbol{y}^m, \boldsymbol{y}_*^l)/\tau_s)} \Big\},
\tag{4}
$$

where $cos(\cdot, \cdot)$ denotes cosine similarity, $\tau_s$ is a temperature parameter, $\boldsymbol{y}^l$ is the generative probability of $\boldsymbol{x}^l$, and $\boldsymbol{y}_*^l$ is the generative probability of $\boldsymbol{x}_*^l$. To align labels across all views, we accumulate contrastive losses as follows:

$$
\mathcal{L}_{con}^{sa} = \sum_{m=1}^M \sum_{l \neq m} \ell_{ml}^{sa}.
\tag{5}
$$

We design another cross-view contrastive loss with cluster-anchor. Denote the $i$-th column of $\boldsymbol{Y}^m \in \mathbb{R}^{N \times |\mathcal{C}_t|}$ as $\boldsymbol{Y}_i^m \in \mathbb{R}^N$, which is the generative probability of the $i$-th cluster to all $N$ samples. Since the generative probability of the same cluster should be consistent across views, the cross-view contrastive loss between $\boldsymbol{Y}^m$ of the $m$-th view and $\boldsymbol{Y}^l$ of the $l$-th view with cluster-anchor is defined as follows:

$$
\ell_{ml}^{ca} = \mathbb{E}_{i \in [|\mathcal{C}_t|]} \Big\{ -log \frac{exp(cos(\boldsymbol{Y}_i^m, \boldsymbol{Y}_i^l)/\tau_c)}{\sum_{j \in [|\mathcal{C}_t|]} exp(cos(\boldsymbol{Y}_i^m, \boldsymbol{Y}_j^l)/\tau_c)} \Big\},
\tag{6}
$$

where $[|\mathcal{C}_t|] = \{1, 2, ..., |\mathcal{C}_t|\}$, and $\tau_c$ is a temperature parameter. Similarly, we accumulate contrastive losses across all views as follows:

$$
\mathcal{L}_{con}^{ca} = \sum_{m=1}^M \sum_{l \neq m} \ell_{ml}^{ca}.
\tag{7}
$$

By optimizing both $\mathcal{L}_{con}^{sa}$ and $\mathcal{L}_{con}^{ca}$, we can align both sample features and cluster features across multiple views to construct the common semantic structure and eliminate view-specific noise in the label space. When predicting the cluster label $L$, we mean-pool the generative probabilities over all views and select the cluster with the highest probability as follows:

$$L = \arg\max_i \left(\frac{1}{M}\sum_{m=1}^{M} y_{(i)}^m\right). \tag{8}$$

### 4.1.3 OPTIMIZATION OF MULTI-VIEW CLUSTER SEARCH

The overall optimization objective of the multi-view cluster search can be written as follows:

$$\mathcal{L}_{MCS} = \mathcal{L}_{rec} + \mathcal{L}_{con}^{sa} + \mathcal{L}_{con}^{ca}. \tag{9}$$

### 4.2 MULTI-VIEW CLUSTER CONSOLIDATION (MCC)

After updating multi-view autoencoders and learning Gaussian components $\Theta_t$ for the $t$-th task in the multi-view cluster consolidation, the learned knowledge of the past $(t-1)$ tasks may be forgotten. Therefore, we conduct a multi-view cluster consolidation to consolidate all learned clusters and prevent catastrophic forgetting.

**Task memory.** In continual learning, the model cannot access the whole past data $\bigcup_{i=1}^{t-1} \mathcal{T}_i$, but can access a small memory $\mathcal{M} = \{s_i = (\boldsymbol{x}_i^1, ..., \boldsymbol{x}_i^M, L_i)\}_{i=1}^{|\mathcal{M}|}$ where $\boldsymbol{x}_i^j$ is the $j$-th view of the $i$-th sample and $L_i$ is its pseudo label. After every MCS phase, we generate pseudo cluster labels by Equation 8 and randomly push $N'$ samples into $\mathcal{M}$, where $N' \ll |\mathcal{T}_t|$ is a hyperparameter. Therefore, $\mathcal{M}$ contains $N'$ samples of every past task 1-$t$, based on which we will conduct our Multi-view Cluster Consolidation.

**Continual component expansion.** Aiming at conducting multi-view clustering on all learned semantic classes $\bigcup_{i=1}^{t} \mathcal{C}_i$, we need to integrate all learned knowledge for the past tasks. Therefore, we continually expand the components of global Gaussian mixture models for solving all past tasks 1-$t$ simultaneously as follows:

$$\Theta \leftarrow \Theta \cup \Theta_t, \tag{10}$$

where $\Theta$ contains all learned Gaussian components, and we can compute the generative probability to all learned clusters by $GMM(\cdot; \Theta)$.

**Self-supervised data replay.** To consolidate all learned clusters and recall knowledge from the task memory, we use the self-supervised pseudo labels to compute the cross-entropy loss as follows:

$$\mathcal{L}_{rpl} = \mathbb{E}_{s_i \in \mathcal{M}}\left\{\frac{1}{M}\sum_{m=1}^{M} CE\big(softmax(GMM(\boldsymbol{h}_i^m; \Theta)), L_i\big)\right\}, \tag{11}$$

where $\boldsymbol{h}_i^m = E^m(\boldsymbol{x}_i^m; \zeta^m)$ is the common feature of $\boldsymbol{x}_i^m$ and $CE(\cdot, \cdot)$ is the cross-entropy function. The objective of $\mathcal{L}_{rpl}$ is to recover the learned multi-view clusters memorized in $\mathcal{M}$. Therefore, by optimizing $\mathcal{L}_{rpl}$ with a small number of past samples, we can efficiently update the model to learn to conduct multi-view clustering on all past semantic classes and recall the learned knowledge for all past tasks 1-$t$.

**Cross-view synchronous loss.** As found in the previous work, multi-view features may have asynchronous convergence rates in learning (Wang et al., 2020; Winterbottom et al., 2020; Peng et al., 2022). Especially in continual multi-view learning, the unbalanced learning rhythm of different views can be amplified by our two-phase and sequential learning manner. Subsequently, features of different views can be misaligned during continual learning, which can impact the effectiveness of mining common semantics across views. Therefore, we propose a cross-view synchronous loss to balance the learning rhythm across different views and obtain aligned multi-view features. Our cross-view synchronous loss between the $m$-th and $l$-th views is based on developing the supervised contrastive loss (Khosla et al., 2020) to cross-view self-supervised contrastive loss as follows:

$$\ell_{ml}^{sy} = \mathbb{E}_{L_i = L_j}\left\{-log\frac{exp(cos(\boldsymbol{h}_i^m, \boldsymbol{h}_j^l)/\tau_v)}{\sum_{s_k \in \mathcal{M}} exp(cos(\boldsymbol{h}_i^m, \boldsymbol{h}_k^l)/\tau_v)}\right\}, \tag{12}$$

where $\tau_v$ is a temperature parameter. We accumulate contrastive losses across all views to obtain the complete cross-view synchronous loss as follows:

$$\mathcal{L}_{syn} = \sum_{m=1}^{M} \sum_{l \neq m} \ell_{ml}^{sy}. \tag{13}$$

By optimizing $\mathcal{L}_{syn}$, we minimize the difference between multi-view features of the same cluster while maximizing the difference between features of different clusters. Therefore, the multi-view features can be updated synchronously in continual learning while the discriminative semantic structure is maintained.

**Optimization.** We also use reconstruction loss $\mathcal{L}_{rec}$ introduced in Equation 2 for optimization in the multi-view cluster consolidation. The overall optimization objective of the multi-view cluster consolidation can be written as follows:

$$\mathcal{L}_{MCC} = \mathcal{L}_{rpl} + \mathcal{L}_{syn} + \mathcal{L}_{rec}. \tag{14}$$

The overall optimization procedure is summarized in Appendix A.

Table 1: The information of the datasets in our experiments.

| Datasets | Training | Test | Views | Classes | Task Classes |
|----------|----------|------|-------|---------|--------------|
| Caltech-2V | 1,120 | 280 | 2 | 7 | [2, 2, 3] |
| Caltech-3V | 1,120 | 280 | 3 | 7 | [2, 2, 3] |
| Caltech-4V | 1,120 | 280 | 4 | 7 | [2, 2, 3] |
| Caltech-5V | 1,120 | 280 | 5 | 7 | [2, 2, 3] |
| MNIST-USPS | 4,000 | 1,000 | 2 | 10 | [2, 2, 2, 2, 2] |
| Fashion | 8,000 | 2,000 | 3 | 10 | [2, 2, 2, 2, 2] |

## 5 EXPERIMENTS

We include more experiments and experimental details in Appendix B. Our code will be made public after acceptance.

### 5.1 DATASETS

Following Xu et al. (2022), we adopt six public datasets in experiments, as shown in Table 1. More details about the datasets are included in Appendix B.2.

### 5.2 BASELINE METHODS AND EVALUATION METRICS

**Baseline methods.** To the best of our knowledge, this paper is the first to study class-incremental continual learning for multi-view clustering. Therefore, we adopt the state-of-the-art MVC methods **MFLVC** (Xu et al., 2022), **DSMVC** (Tang & Liu, 2022), and **JCT** (Hu et al., 2023) and continual learning methods **EWC** (Kirkpatrick et al., 2017), **LwF** (Li & Hoiem, 2016), and **CVS** (Wan et al., 2022a). To construct more strong baselines for continual MVC, we further combine MVC methods with continual learning strategies EWC and **data replay**, where data replay utilizes the same task memory in our CCMVC and replays buffered data for MVC models after learning each task. Additionally, **Joint Training** is trained from scratch on all past training sets jointly, using the model and loss as the same as our multi-view cluster search. We note that Joint Training is **not with continual learning setting**, but acts as the performance **upper bound** of continual learning methods. The details of baseline methods are included in Appendix B.3.

**Evaluation metrics.** When complete learning the $t$-th task on the training set $\mathcal{T}_t$, we evaluate the model on the test set $\widetilde{\mathcal{T}_t}$ for all past tasks, which contains all learned classes $\bigcup_{i=1}^{t} \mathcal{C}_i$. We report the mean accuracy $A_t$ (aka, **Class Incremental Accuracy**) on $\widetilde{\mathcal{T}_t}$ for all $t$ of 10 runs.

### 5.3 RESULTS AND DISCUSSIONS

The clustering results of all compared methods on six datasets are shown in Tables 2-3. Based on these results, we have the following observations: (1) Our proposed CCMVC outperforms all baseline methods with clear margins on all datasets for class-incremental continual learning. Compared

Table 2: Results of all methods on Caltech dataset with different views. $A_i$ denotes the class incremental accuracy on past $i$ tasks. Bold denotes the best results and underline denotes the second-best.

| Methods | Caltech-2V | | | Caltech-3V | | | Caltech-4V | | | Caltech-5V | | |
|---|---|---|---|---|---|---|---|---|---|---|---|---|
| | $A_1$ | $A_2$ | $A_3$ | $A_1$ | $A_2$ | $A_3$ | $A_1$ | $A_2$ | $A_3$ | $A_1$ | $A_2$ | $A_3$ |
| Joint Training | 0.925 | 0.931 | 0.660 | 0.987 | 0.968 | 0.681 | 1.000 | 0.981 | 0.817 | 1.000 | 0.993 | 0.835 |
| EWC | 0.575 | 0.543 | 0.385 | 0.650 | 0.456 | 0.378 | 0.625 | 0.437 | 0.321 | 0.562 | 0.493 | 0.389 |
| LwF | 0.825 | 0.500 | 0.342 | 0.885 | 0.612 | 0.401 | 0.912 | 0.672 | 0.437 | 0.925 | 0.695 | 0.434 |
| CVS | 0.825 | 0.452 | 0.298 | 0.885 | 0.601 | 0.438 | 0.912 | 0.670 | 0.458 | 0.925 | 0.721 | 0.490 |
| MFLVC | 0.825 | 0.456 | 0.282 | 0.885 | 0.650 | 0.403 | 0.912 | 0.675 | 0.422 | 0.925 | 0.708 | 0.451 |
| MFLVC+EWC | 0.825 | 0.531 | 0.385 | 0.885 | 0.600 | 0.425 | 0.912 | 0.712 | 0.464 | 0.925 | 0.721 | 0.478 |
| MFLVC+replay | 0.825 | 0.538 | 0.400 | 0.885 | 0.582 | 0.420 | 0.912 | 0.702 | 0.480 | 0.925 | 0.728 | 0.504 |
| MFLVC+EWC+replay | 0.825 | 0.556 | 0.410 | 0.885 | 0.618 | 0.431 | 0.912 | 0.731 | 0.486 | 0.925 | 0.741 | 0.513 |
| DSMVC | 0.800 | 0.507 | 0.257 | 0.850 | 0.487 | 0.418 | 0.875 | 0.597 | 0.412 | 0.898 | 0.633 | 0.446 |
| DSMVC+EWC | 0.800 | 0.523 | 0.308 | 0.850 | 0.505 | 0.437 | 0.875 | 0.631 | 0.452 | 0.898 | 0.661 | 0.510 |
| DSMVC+replay | 0.800 | 0.543 | 0.365 | 0.850 | 0.566 | 0.472 | 0.875 | 0.637 | 0.481 | 0.898 | 0.653 | 0.549 |
| DSMVC+EWC+replay | 0.800 | 0.588 | 0.424 | 0.850 | 0.602 | 0.461 | 0.875 | 0.655 | 0.504 | 0.898 | 0.683 | 0.560 |
| JCT | 0.825 | 0.531 | 0.245 | 0.862 | 0.500 | 0.398 | 0.892 | 0.662 | 0.457 | 0.940 | 0.686 | 0.462 |
| JCT+EWC | 0.825 | 0.538 | 0.305 | 0.862 | 0.516 | 0.458 | 0.892 | 0.689 | 0.471 | 0.940 | 0.693 | 0.507 |
| JCT+replay | 0.825 | 0.543 | 0.335 | 0.862 | 0.506 | 0.431 | 0.892 | 0.637 | 0.481 | 0.940 | 0.676 | 0.499 |
| JCT+EWC+replay | 0.825 | 0.601 | 0.441 | 0.862 | 0.668 | 0.488 | 0.892 | 0.677 | 0.514 | 0.940 | 0.793 | 0.581 |
| **CCMVC (Ours)** | **0.887** | **0.768** | **0.489** | **0.925** | **0.807** | **0.557** | **0.935** | **0.832** | **0.585** | **0.950** | **0.850** | **0.645** |

Table 3: Results of all methods on two datasets. $A_i$ denotes the class incremental accuracy on past $i$ tasks. Bold denotes the best results and underline denotes the second-best.

| Methods | Fashion | | | | | MNIST-USPS | | | | |
|---|---|---|---|---|---|---|---|---|---|---|
| | $A_1$ | $A_2$ | $A_3$ | $A_4$ | $A_5$ | $A_1$ | $A_2$ | $A_3$ | $A_4$ | $A_5$ |
| Joint Training | 1.000 | 1.000 | 0.998 | 0.965 | 0.995 | 1.000 | 1.000 | 0.998 | 0.998 | 0.997 |
| EWC | 0.912 | 0.690 | 0.447 | 0.419 | 0.340 | **1.000** | 0.615 | 0.528 | 0.382 | 0.363 |
| LwF | **1.000** | 0.700 | 0.709 | 0.552 | 0.445 | **1.000** | 0.702 | 0.640 | 0.550 | 0.442 |
| CVS | **1.000** | 0.702 | 0.476 | 0.344 | 0.300 | **1.000** | 0.695 | 0.567 | 0.502 | 0.412 |
| MFLVC | **1.000** | 0.498 | 0.493 | 0.267 | 0.286 | **1.000** | 0.687 | 0.500 | 0.380 | 0.323 |
| MFLVC+EWC | **1.000** | 0.561 | 0.525 | 0.348 | 0.305 | **1.000** | 0.710 | 0.556 | 0.428 | 0.366 |
| MFLVC+replay | **1.000** | 0.546 | 0.503 | 0.356 | 0.299 | **1.000** | 0.688 | 0.531 | 0.398 | 0.352 |
| MFLVC+EWC+replay | **1.000** | 0.572 | 0.549 | 0.370 | 0.334 | **1.000** | 0.710 | 0.601 | 0.490 | 0.405 |
| DSMVC | 0.985 | 0.636 | 0.564 | 0.398 | 0.348 | 0.995 | 0.622 | 0.492 | 0.387 | 0.348 |
| DSMVC+EWC | 0.985 | 0.672 | 0.569 | 0.417 | 0.441 | 0.995 | 0.663 | 0.509 | 0.466 | 0.365 |
| DSMVC+replay | 0.985 | 0.656 | 0.587 | 0.507 | 0.458 | 0.995 | 0.650 | 0.535 | 0.488 | 0.397 |
| DSMVC+EWC+replay | 0.985 | 0.674 | 0.596 | 0.573 | 0.482 | 0.995 | 0.684 | 0.571 | 0.511 | 0.404 |
| JCT | **1.000** | 0.683 | 0.570 | 0.411 | 0.299 | **1.000** | 0.690 | 0.530 | 0.442 | 0.293 |
| JCT+EWC | **1.000** | 0.701 | 0.595 | 0.460 | 0.352 | **1.000** | 0.710 | 0.546 | 0.458 | 0.381 |
| JCT+replay | **1.000** | 0.695 | 0.583 | 0.440 | 0.510 | **1.000** | 0.700 | 0.596 | 0.478 | 0.397 |
| JCT+EWC+replay | **1.000** | 0.748 | 0.686 | 0.590 | 0.535 | **1.000** | 0.721 | 0.625 | 0.535 | 0.458 |
| **CCMVC (Ours)** | **1.000** | **0.965** | **0.923** | **0.833** | **0.721** | **1.000** | **0.957** | **0.885** | **0.776** | **0.729** |

with the second-best results, our CCMVC improves the class incremental accuracy of the last task by 0.089, 0.112, 0.098, 0.112, 0.358, and 0.214 on Caltech-2V, Caltech-3V, Caltech-4V, Caltech-5V, Fashion, and MNIST-USPS, respectively. These results demonstrate that the proposed CCMVC can better overcome the catastrophic forgetting problem in continual learning and perform continual multi-view clustering. (2) The performance of the state-of-the-art MVC method (i.e., MFLVC) severely drops after learning more tasks, which suggests that class-incremental continual learning is a challenging problem and cannot be handled well by traditional MVC methods. Moreover, the results of MFLVC integrated with different CL strategies show that naively combining MVC methods and CL methods is sub-optimal for solving the CL problem for MVC. (3) The state-of-the-art method for continual learning (i.e., CVS) performs badly on the adopted multi-view datasets, which may be due to the neglect of view heterogeneity. Different from single-view learning, MVC requires overcoming heterogeneity and discovering the common clustering structure by learning from all available views. These results also indicate that our proposed problem (i.e., class-incremental continual learning) is not a trivial problem in the field of continual learning.

## 5.4 ABLATION STUDY

To further investigate the effectiveness of the proposed losses in Equations 9 and 14, we design several variants by removing one of all loss terms to conduct an ablation study. Table 4 shows the experimental results on two datasets, from which we have the following observations: (1) The performance of our method declines when removing any one of all loss terms, which demonstrates every loss component contributes to the final results. (2) The variant w/o $\mathcal{L}_{rec}$ achieves low scores on two datasets. The reconstruction loss $\mathcal{L}_{rec}$ can prevent the view-related autoencoder from collapsing, which is a basis of multi-view feature learning. (3) The variants w/o $\mathcal{L}_{con}^{sa}$ or $\mathcal{L}_{con}^{ca}$ perform

Table 4: Ablation study of loss components on Fashion and Caltech-5V datasets.

| Variants | Fashion | | | | | Caltech-5V | | |
|---|---|---|---|---|---|---|---|---|
| | $A_1$ | $A_2$ | $A_3$ | $A_4$ | $A_5$ | $A_1$ | $A_2$ | $A_3$ |
| w/o $\mathcal{L}_{rec}$ | **1.000** | 0.746 | 0.664 | 0.467 | 0.393 | 0.915 | 0.743 | 0.550 |
| w/o $\mathcal{L}_{con}^{sa}$ | **1.000** | 0.950 | 0.828 | 0.656 | 0.624 | 0.937 | 0.731 | 0.557 |
| w/o $\mathcal{L}_{con}^{ca}$ | **1.000** | 0.943 | 0.825 | 0.698 | 0.637 | 0.562 | 0.525 | 0.396 |
| w/o $\mathcal{L}_{rpl}$ | **1.000** | 0.931 | 0.784 | 0.656 | 0.539 | **0.950** | 0.681 | 0.560 |
| w/o $\mathcal{L}_{syn}$ | **1.000** | 0.943 | 0.822 | 0.734 | 0.628 | **0.950** | 0.618 | 0.489 |
| **CCMVC** | **1.000** | **0.965** | **0.923** | **0.833** | **0.721** | **0.950** | **0.850** | **0.645** |

badly on two datasets. Different from the traditional EM algorithm for GMM with single-view data, we propose cross-view contrastive learning to optimize GMM. (4) The variant w/o $\mathcal{L}_{rpl}$ performs significantly worse than CCMVC. By replaying the buffered data with pseudo labels, $\mathcal{L}_{rpl}$ can optimize the model to recover clusters memorized in the task memory and prevent catastrophic forgetting. (5) The variant w/o $\mathcal{L}_{syn}$ performs much worse than CCMVC. By developing cross-view self-supervised contrastive learning, we minimize the difference of features in the same clusters across views, which mitigates the asynchronous learning rhythm of different views.

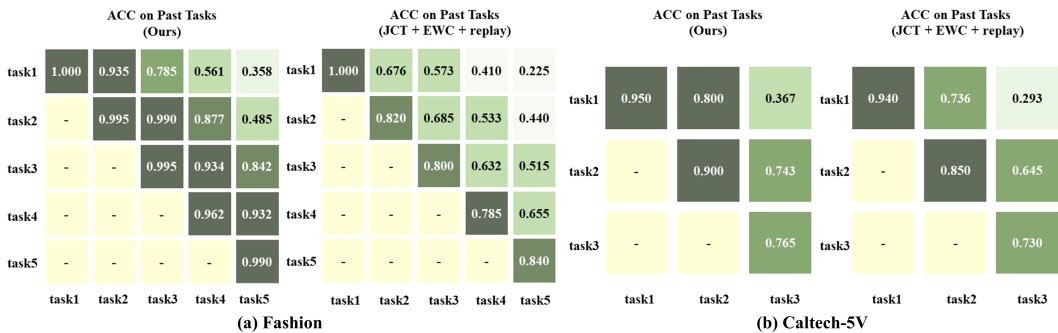

Figure 3: Performance of Different Tasks in the continual learning process on Fashion and Caltech-5V. The x-axis denotes the sequential learning process and the y-axis denotes the per-task accuracy.

## 5.5 PERFORMANCE OF DIFFERENT TASKS IN CONTINUAL LEARNING PROCESS

To further investigate the performance of different tasks in continual learning, we show the heatmaps of per-task accuracy on Fashion and Caltach-5V in Figure 3. In every heat map, the x-axis denotes the sequential learning process in continual learning and the y-axis denotes the per-task accuracy while clustering for all past tasks simultaneously. Comparing our CCMVC with the strongest baseline JCT+EWC+replay, we can find that our CCMVC performs better, if not the same, for all individual tasks during the continual learning process, especially for the past tasks. These results further demonstrate the superiority of our proposed method for alleviating forgetting the knowledge of past tasks in continual learning and continually conducting clustering for streaming multi-view data. Moreover, we can also find that the earlier the task, the worse the performance in the whole process for both compared methods. Therefore, the knowledge-forgetting problem in continual learning is still not completely solved, which is left as a challenge of class-incremental learning for MVC.

## 6 CONCLUSION

In this paper, we explore class-incremental continual learning for multi-view clustering for the first time and propose a Class-incremental Continual Multi-View Clustering (CCMVC) model. To prevent catastrophic forgetting while learning new semantic classes, we propose a continual component expansion with self-supervised data replay to continually learn incremental clustering components. To balance the learning rhythm across different views during continual learning, we propose a cross-view synchronous loss by developing cross-view self-supervised contrastive learning. Extensive experiments conducted on six public benchmark datasets indicate that CCMVC significantly outperforms state-of-the-art approaches.

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

## A  OPTIMIZATION PROCEDURE

The overall optimization procedure is demonstrated in Algorithm 1.

---

**Algorithm 1** Optimization procedure of our proposed CCMVC

---

**Input:** Sequential training task set $\{\mathcal{T}_1, \mathcal{T}_2, ...\}$; hyper-parameters $D, N', \tau_s, \tau_c, \tau_v$; batch size $bs$; learning rate $\xi$.
**Output:** Optimized parameters $\zeta, \phi$ of view-related autoencoders, $\Theta$ of Gaussian mixture models.

1: Initialize $t=0$;
2: **while** the $t$-th task comes with training set $\mathcal{T}_t$ **do**
3:     Randomly initialize Gaussian components $\Theta_t$.
4:     **repeat**
5:         **for** $\lfloor \frac{|\mathcal{T}_t|}{bs} \rfloor$ iterations **do**
6:             Randomly select $bs$ samples in $\mathcal{T}_t$ to form a batch.
7:             Obtain the multi-view features by Equation 1 for the batch.
8:             Calculate the loss of **multi-view cluster search** in Equation 9.
9:             Back propagate gradients and update model parameters:
            $(\zeta, \phi, \Theta_t) \leftarrow (\zeta, \phi, \Theta_t) - \xi \frac{\partial}{\partial(\zeta, \phi, \Theta_t)} \mathcal{L}_{MCS}$.
10:         **end for**
11:     **until** convergence;
12:     Generate pseudo labels for $\mathcal{T}_t$ by Equation 8.
13:     Randomly add $N'$ samples with pseudo labels in $\mathcal{T}_t$ into task memory $\mathcal{M}$.
14:     Expand the global Gaussian mixture models: $\Theta \leftarrow \Theta \cup \Theta_t$.
15:     **repeat**
16:         **for** $\lfloor \frac{|\mathcal{M}|}{bs} \rfloor$ iterations **do**
17:             Randomly select $bs$ samples in $\mathcal{M}$ to form a batch.
18:             Obtain the multi-view features by Equation 1 for the batch.
19:             Calculate the loss of **multi-view cluster consolidation** in Equation 14.
20:             Back propagate gradients and update model parameters:
            $(\zeta, \phi, \Theta) \leftarrow (\zeta, \phi, \Theta) - \xi \frac{\partial}{\partial(\zeta, \phi, \Theta)} \mathcal{L}_{MCC}$.
21:         **end for**
22:     **until** convergence;
23:     $t \leftarrow t + 1$.
24: **end while**

---

## B  EXPERIMENTS

### B.1  IMPLEMENTATION DETAILS

Our code is implemented by PyTorch (Paszke et al., 2019) and ran on one RTX 3090 GPU. Adam optimizer (Kingma & Ba, 2015) is adopted for optimization with a learning rate of 3e-4. The batch size is set as 256 for Fashion and MNIST-USPS and 128 for Caltech. We set the dimension of common feature space as $D = 512$, following MFLVC (Xu et al., 2022). The hidden sizes of each view-related encoder $E^m$ are 500, 500, 2000, and 512. The hidden sizes of each view-related decoder $D^m$ are 2000, 500, 2000, and the dimension of the input data. All layers except the last layer of encoders and decoders are activated by ReLU function. The buffer size $N'$ for the $t$-th task is set as $5\%|\mathcal{T}_t|$ on all datasets. For contrastive learning, we set $\tau_s = 4.3$ and $\tau_c = 7$ on Fashion, $\tau_s = 0.4$ and $\tau_c = 0.5$ on MNIST-USPS, $\tau_s = 1$ and $\tau_c = 1$ on Caltech. $\tau_v$ is set as 1 for all datasets. We follow previous work of multiview clustering (Zhou & Shen, 2020; Trosten et al., 2021; Xu et al., 2022) to add all losses of our model without trade-off parameters. The same training setting is applied to all compared methods to conduct a fair comparison.

## B.2 DATASETS

Six public datasets are adopted in experiments. **Caltech** (Fei-Fei et al., 2004) is an RGB image dataset with multiple views, based on which four datasets with different numbers of views are built following Xu et al. (2022). Concretely, **Caltech-2V** includes *WM* and *CENTRIST*; **Caltech-3V** includes *WM*, *CENTRIST*, and *LBP*; **Caltech-4V** includes *WM*, *CENTRIST*, *LBP*, and *GIST*; **Caltech-5V** includes *WM*, *CENTRIST*, *LBP*, *GIST* and *HOG*. We manually divide Caltech into 3 tasks with 2, 2, and 3 classes, respectively. **MNIST-USPS** (LeCun et al., 1998) contains digital images with two different styles. MNIST-USPS are divided into 5 tasks and each task comprises 2 classes. **Fashion** (Xiao et al., 2017) is an image dataset about products, where we follow Xu et al. (2021) to treat different three styles as three views. Fashion is also divided into 5 tasks and each task comprises 2 classes. All datasets are divided into training sets and test sets with a proportion of 4:1.

## B.3 BASELINES

The details of baseline methods are listed as follows:

**Joint Training** is trained from scratch on all past training sets jointly, using the model and loss as the same as our plastic stage. We note that Joint Training is **not** with continual learning setting, but acts as the performance **upper bound** of continual learning methods.

**MFLVC (Xu et al., 2022)** is the state-of-the-art method for multi-view clustering, which learns different levels of features from the raw features in a fusion-free manner. We adapt MFLVC to our class-incremental continual learning setting.

**DSMVC (Tang & Liu, 2022)** is also a state-of-the-art model for multi-view clustering, which is trained to simultaneously extract complementary information and discard the meaningless noise by automatically selecting features.

**JCT (Hu et al., 2023)** is a state-of-the-art model for multi-view clustering, which combines feature-level alignment-oriented and commonality-oriented contrastive learning, and cluster-level consistency-oriented contrastive learning.

**EWC (Kirkpatrick et al., 2017)** remembers old tasks by selectively slowing down learning on the weights important for those tasks to mitigate catastrophic forgetting in artificial neural networks.

**LwF (Li & Hoiem, 2016)** exploits knowledge distillation to retrain representation for previous tasks and uses L2 distance as the regularization term to mitigate catastrophic forgetting.

**CVS (Wan et al., 2022a)** proposes losses for inter-task data coherence, neighbor-task model coherence, and intra-task discrimination to improve continual learning.

**MFLVC/DSMVC/JCT+EWC** integrates the multi-view clustering module of MFLVC/DSMVC/JCT and the continual learning module of EWC to construct a continual MVC baseline.

**MFLVC/DSMVC/JCT+replay** utilizes the same task memory in our CCMVC and replays buffered data for MFLVC/DSMVC/JCT after learning each task.

**MFLVC/DSMVC/JCT+EWC+replay** integrates the multi-view clustering module of MFLVC/DSMVC/JCT, the continual learning module of EWC, and data replay to construct a stronger continual MVC baseline.

We adopt the officially released code to reimplement all baselines.

## B.4 ANALYSIS OF CLASS INCREMENTAL ORDER

By default, we utilize the default class order of every dataset, and the classes in the test are incremented from the first to the last class. In this section, we analyze the effect of class incremental order in continual learning for multi-view clustering. We change the order of the class increments on Fashion dataset and show the results in Tables 5-7. In these tables, $c_0, ..., c_9$ denote 10 classes with the default class order in Fashion. "Task Classes" denotes the classes in the training set of the divided tasks. In the $i$-th test, the test data are from the classes of all past $i$ tasks. From the results, we have the following observations:

Table 5: Results on Fashion with different class incremental order. $A_i$ denotes the class incremental accuracy on all past $i$ tasks. Bold denotes the best results and underline denotes the second-best.

| Task Classes | $c_4, c_5$ | $c_0, c_1$ | $c_2, c_3$ | $c_8, c_9$ | $c_6, c_7$ |
|---|---|---|---|---|---|
| Methods | $A_1$ | $A_2$ | $A_3$ | $A_4$ | $A_5$ |
| LwF | **1.000** | 0.681 | 0.601 | 0.501 | 0.425 |
| CVS | **1.000** | 0.644 | 0.500 | 0.433 | 0.302 |
| DSMVC | 0.965 | 0.592 | 0.513 | 0.367 | 0.345 |
| DSMVC+EWC | 0.965 | 0.640 | 0.541 | 0.451 | 0.405 |
| DSMVC+replay | 0.965 | 0.668 | 0.570 | 0.477 | 0.435 |
| DSMVC+EWC+replay | 0.965 | 0.692 | 0.614 | 0.531 | 0.475 |
| JCT | **1.000** | 0.635 | 0.574 | 0.429 | 0.305 |
| JCT+EWC | **1.000** | 0.682 | 0.592 | 0.472 | 0.341 |
| JCT+replay | **1.000** | 0.688 | 0.608 | 0.464 | 0.441 |
| JCT+EWC+replay | **1.000** | 0.712 | 0.648 | 0.569 | 0.522 |
| **CCMVC (Ours)** | **1.000** | **0.916** | **0.848** | **0.790** | **0.627** |

Table 6: Results on Fashion with different class incremental order. $A_i$ denotes the class incremental accuracy on all past $i$ tasks. Bold denotes the best results and underline denotes the second-best.

| Task Classes | $c_5, c_7$ | $c_0, c_3$ | $c_1, c_2$ | $c_8, c_9$ | $c_4, c_6$ |
|---|---|---|---|---|---|
| Methods | $A_1$ | $A_2$ | $A_3$ | $A_4$ | $A_5$ |
| LwF | **1.000** | 0.690 | 0.595 | 0.500 | 0.400 |
| CVS | **1.000** | 0.678 | 0.518 | 0.421 | 0.322 |
| DSMVC | 0.970 | 0.612 | 0.505 | 0.350 | 0.347 |
| DSMVC+EWC | 0.970 | 0.665 | 0.537 | 0.440 | 0.421 |
| DSMVC+replay | 0.970 | 0.670 | 0.572 | 0.465 | 0.432 |
| DSMVC+EWC+replay | 0.970 | 0.702 | 0.602 | 0.524 | 0.445 |
| JCT | **1.000** | 0.639 | 0.565 | 0.417 | 0.325 |
| JCT+EWC | **1.000** | 0.698 | 0.588 | 0.437 | 0.346 |
| JCT+replay | **1.000** | 0.705 | 0.601 | 0.440 | 0.450 |
| JCT+EWC+replay | **1.000** | 0.721 | 0.640 | 0.551 | 0.525 |
| **CCMVC (Ours)** | **1.000** | **0.941** | **0.814** | **0.721** | **0.646** |

Table 7: Results on Fashion with different class incremental order. $A_i$ denotes the class incremental accuracy on all past $i$ tasks. Bold denotes the best results and underline denotes the second-best.

| Task Classes | $c_4, c_7$ | $c_0, c_1$ | $c_2, c_3$ | $c_5, c_9$ | $c_6, c_8$ |
|---|---|---|---|---|---|
| Methods | $A_1$ | $A_2$ | $A_3$ | $A_4$ | $A_5$ |
| LwF | **1.000** | 0.675 | 0.592 | 0.481 | 0.417 |
| CVS | **1.000** | 0.643 | 0.519 | 0.441 | 0.330 |
| DSMVC | 0.932 | 0.590 | 0.517 | 0.360 | 0.355 |
| DSMVC+EWC | 0.932 | 0.635 | 0.543 | 0.446 | 0.428 |
| DSMVC+replay | 0.932 | 0.667 | 0.582 | 0.467 | 0.435 |
| DSMVC+EWC+replay | 0.932 | 0.691 | 0.610 | 0.541 | 0.481 |
| JCT | **1.000** | 0.640 | 0.582 | 0.419 | 0.335 |
| JCT+EWC | **1.000** | 0.705 | 0.600 | 0.472 | 0.352 |
| JCT+replay | **1.000** | 0.695 | 0.612 | 0.451 | 0.451 |
| JCT+EWC+replay | **1.000** | 0.718 | 0.668 | 0.554 | 0.531 |
| **CCMVC (Ours)** | **1.000** | **0.945** | **0.860** | **0.756** | **0.675** |

- With different class incremental orders, the performance $A_i$ of the compared methods is also different. This indicates that the class incremental order can affect the model performance. Therefore, we compare different methods with the same class incremental order in our experiments to ensure fairness.

- With different class incremental orders, the performance $A_i$ of our proposed CCMVC consistently outperforms all baselines especially when more classes are learned. These results further demonstrate the effectiveness of our method for continual multi-view clustering.

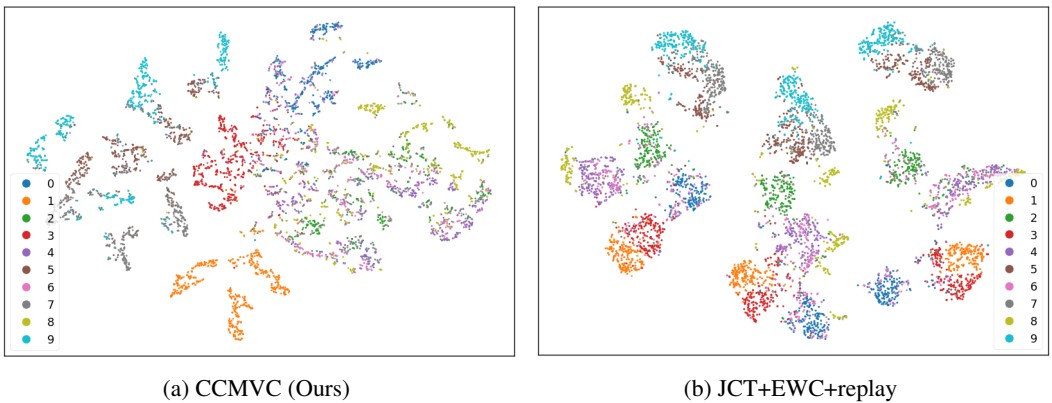

(a) CCMVC (Ours)                                     (b) JCT+EWC+replay

Figure 4: Cluster visualization of the learned multi-view features belonging to different classes on Fashion dataset.

### B.5    CLUSTER VISUALIZATION

To further investigate the effectiveness of the learned multi-view features, we visualize the learned features of CCMVC and JCT+ EWC+replay after learning from all tasks on Fashion to conduct a comparison. Learned features of all views are visualized via t-SNE algorithm (Van der Maaten & Hinton, 2008) and dyed with different colors to represent different ground truth classes. The visualization results are shown in Figure 4, and we have the following observations: The feature clusters of CCMVC are more compact than the clusters of JCT+EWC+replay. For example, the features of class 1 or 3 learned by CCMVC are almost in a single cluster, while these features learned by JCT+EWC+replay are divided and mixed into three clusters. Generally, the small clusters in Figure 4(a) are more likely to be of a single class than clusters in Figure 4(b). The visualization results further demonstrate that our CCMVC can better learn multi-view features for continually clustering multi-view data.

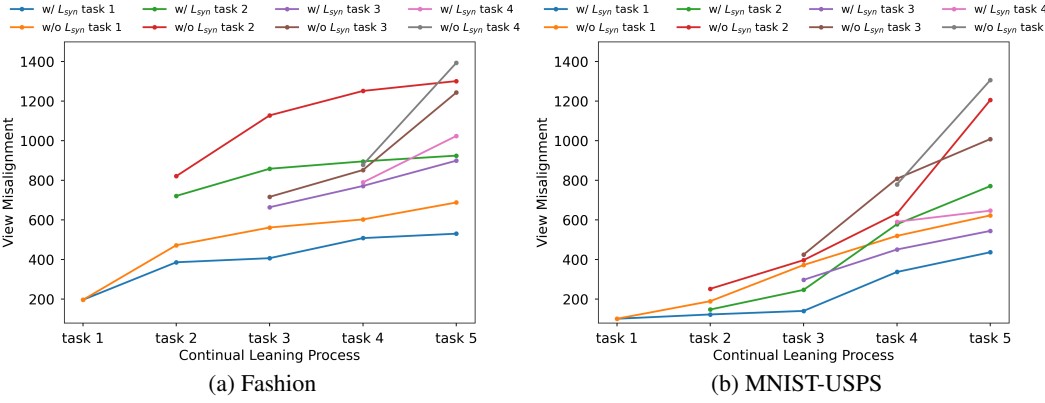

(a) Fashion                                     (b) MNIST-USPS

Figure 5: View asynchronicity of the learned multi-view features belonging to different tasks during the continual learning process on Fashion and MNIST-USPS datasets.

## B.6 ANALYSIS OF VIEW ASYNCHRONICITY

As discussed in Section 1, the asynchronicity across different views is an important challenge in multi-view continual learning. Therefore, in this section, we empirically analyze the view asynchronicity in the continual learning process. Specifically, we compute the mean Euclidean distance among multi-view features of the same test sample to represent the view misalignment. The mean view misalignment of the test data belonging to specific tasks during the continual learning process is shown in Figure 5. We compare the results of our CCMVC (w/ $\mathcal{L}_{syn}$) and the results of the variant w/o $\mathcal{L}_{syn}$ on Fashion and MNIST-USPS datasets. From the experimental results, we can observe that the view misalignment of all tasks is significantly mitigated on both datasets when applying our proposed cross-view synchronous loss $\mathcal{L}_{syn}$. The proposed $\mathcal{L}_{syn}$ balances the learning rhythm across different views and aligns multi-view features during the data replay. Therefore, the multi-view features can be updated synchronously in continual learning while the discriminative semantic structure is maintained.

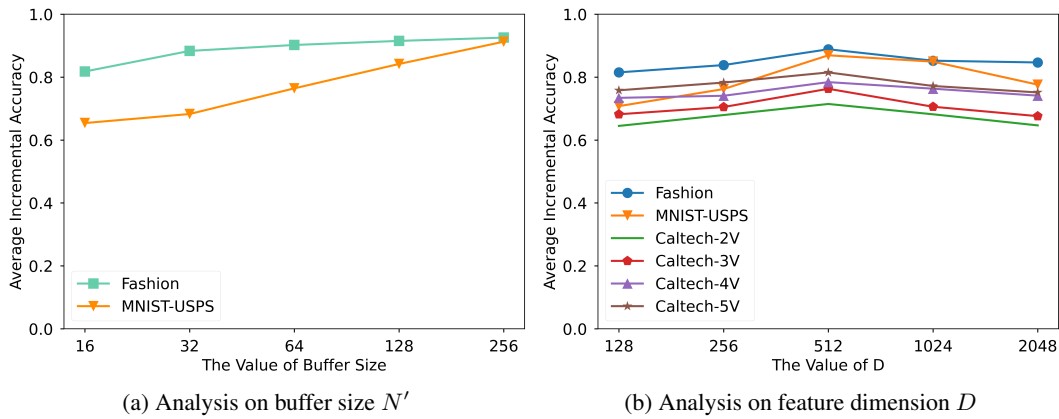

(a) Analysis on buffer size $N'$          (b) Analysis on feature dimension $D$

Figure 6: Analysis of hyperparameter sensitivity on different datasets. Average Incremental Accuracy denotes the mean of all Class-incremental Accuracy $A_i$.

## B.7 HYPERPARAMETER SENSITIVITY

In this section, we analyze the sensitivity of the hyper-parameters in our proposed model.

**Buffer Size $N'$**   We explore the sensitivity of buffer size $N'$ in the task. We change the value of $N'$ from 16 to 256 and report the average $A_t$ on Fashion and MNIST-USPS in Figure 6(a). We can observe that when introducing more memory space, the average accuracy increases since more data of the learned clusters can be replayed in the stable stage. However, since storing and replaying past data is expensive in continual learning, an appropriate $N'$ should be selected according to practical conditions.

**Feature Dimension $D$**   In order to estimate the sensitivity of feature dimension $D$, we vary $D$ from 128 to 2048 to evaluate the model performance on six datasets. Figure 6(b) demonstrates the results, and we can observe that the performance of the model is poor with the lower dimension because of the insufficient representation ability, and will decrease with the higher dimension due to the over-fitting problem. By selecting the appropriate dimension, the model can achieve the best performance.

