# OpenReview forum: "Class-Incremental Continual Learning for Multi-View Clustering"
_ICLR.cc/2024/Conference — Submitted to ICLR 2024_

### Official Review · Reviewer_6odo · 2023-10-28

**Soundness:** 3 good
**Presentation:** 2 fair
**Contribution:** 2 fair
**Rating:** 3
**Confidence:** 4

**Summary:**

This paper focuses on the practical problem of streaming multi-view data clustering and propose a Class-incremental Continual Multi-View Clustering (CCMVC) method, which can continually learn incremental clusters for streaming multi-view data and memorize learned knowledge to perform all learned clustering tasks simultaneously. Experimental results on six MVC datasets demonstrate a significant performance improvement of CCMVC compared with the state-of-the-art methods.

**Strengths:**

1.	This paper proposes a multi-view clustering method for streaming multi-view data with incremental classes, which is frequent in real-world applications.
2.	The authors provide a detailed explanation of the algorithm, and the experimental design and analysis are also comprehensive.

**Weaknesses:**

1.	The proposed method only learns the common feature sapce, and ignores the contribution of heterogeneity among multiple views to the clustering.
2.	Although the experimental results demonstrate the effectiveness of the proposed method, the two-stage independent optimization approach carries the risk of inconsistent optimization.
3.	The paper lacks a computational complexity analysis. With an increasing number of views, is the clustering performance and the computational complexity of the proposed method stable?

**Questions:**

According to Eq. (11), the pseudo label set L is important to data replay and continuous clustering. If the quality of L is low, will it affect the performance of both the MCS and MCC modules?
	Eq. (12) requires calculating the similarity of features between pairs of views. Does this significantly reduce the computational efficiency of the proposed method?
	There are many important hyperparameters in the proposed method, however, the authors did not explain how to select the suitable parameter value, nor did they conduct relevant parameter analysis experiments. In particular, the size of task memory, N', can significantly affect the performance of the MCC module.
	In Tables 2, 3 and 4, authors should provide the variance or standard deviation of each result and further verify the statistical significance of these results. In addition, the authors can supplement the dynamic visualization results of CCMVC on different datasets to visually verify its performance on streaming multi-view data.

---

### Official Review · Reviewer_krpX · 2023-10-29

**Soundness:** 2 fair
**Presentation:** 3 good
**Contribution:** 2 fair
**Rating:** 3
**Confidence:** 4

**Summary:**

This paper discusses multi-view clustering (MVC) in a class-incremental continual learning setting, where multi-view data with incremental semantic classes come sequentially. The paper claims that (1) existing MVC methods focus on learning from static training data and (2) existing continual learning methods only consider single-view data. This paper delineates a method, termed as CCMVC, to handle class-incremental continual learning for multi-view clustering.  Experimental results on six datasets show its promising performance against baseline methods.

**Strengths:**

S1. The paper is well-structured and clearly presented on its motivation as stated in its title, class-incremental continual learning for multi-view clustering.

S2. The originality lies in the two-stage learning with a synergistic integration of view-related autoencoders, cross-view contrastive learning for GMM, and data replay, addressing the distinct challenges posed by class-incremental multi-view clustering.

S3. The main novelty is underscored by dual-anchor contrastive learning and self-supervised data replay with pseudo labels.

**Weaknesses:**

W1. Although the paper provides a well-structured exposition of the proposed methodology, further technical insights regarding the implementation and specific algorithms within the CCMVC method would be beneficial.

W2. The manuscript could delve deeper into the significance of class-incremental multi-view clustering for real-world applications, a factor which could be potential to inspire future research in this domain.

W3. The paper falls short in providing a detailed analysis of the limitations of the proposed approach, a factor which could be significant for future research and practical applications.

W4. The computational complexity of the proposed algorithm, which could be a concern for large-scale datasets or real-world applications, is not discussed in the manuscript.

W5. A more detailed exposition of the datasets used in the evaluation, including their characteristics and potential biases, would enrich the manuscript.

**Questions:**

C1. What are the semantics of multi-view data? How does the method mine them?

C2. Missing related work, e.g., MVStream: Multi-View Data Stream Clustering (TNNLS, 2020), Split-merge Evolutionary Clustering for Multi-view Streaming Data (Procedia Computer Science, 2020), Incremental Multi-view Spectral Clustering (KBS, 2019).

C3. Why and how does Eq. (2) avoid model collapse?

C4. How does the model update each component in GMM? Does the number of components increase as the number of classes increases?

C5. Does the random selection in task memory have performance biases?

C6. Those hyperparameters somewhat degrade the significance of the proposed method for real-world applications.

---

### Official Review · Reviewer_6hBX · 2023-11-01

**Soundness:** 1 poor
**Presentation:** 2 fair
**Contribution:** 2 fair
**Rating:** 3
**Confidence:** 5

**Summary:**

This paper addresses multi-view clustering and proposes a continual learning method for solving class-incremental problems. The proposed method utilizes self-supervised data replay to learn incremental clustering components and prevent forgetting.

**Strengths:**

1. The idea of utilizing self-supervised data replay for learning incremental clustering components and preventing forgetting is interesting.
2. The authors include ablation studies where they remove individual losses to further investigate the proposed method.

**Weaknesses:**

1. The main concern is regarding the experiment. The dataset used is relatively small (1000~10000 samples with 7~10 classes) to effectively validate the effectiveness of the proposed method. Considering the problem revolves around increasing classes, the inclusion of only about 2 new classes may not sufficiently demonstrate the ability of the proposed method in handling new classes.
2. The approach of cross-view contrastive learning seems not novel to the community, as it is largely based on contrastive clustering methods with the only difference being the contrast of clustering anchors.
2. The complexity of the proposed method is not adequately discussed. It would be helpful to compare the computation cost of the proposed method to the baselines.
3. The choice of using GMM for clustering is not justified. It would be beneficial if the authors provided a reason for this selection or discussed alternative methods.

**Questions:**

See the weaknesses.

---

### Official Review · Reviewer_UeiP · 2023-11-04

**Soundness:** 2 fair
**Presentation:** 2 fair
**Contribution:** 2 fair
**Rating:** 5
**Confidence:** 5

**Summary:**

This paper focuses on incremental learning in multi-view clustering. Existing MVC methods have trouble dealing with streaming multi-view data with incremental classes, and the proposed method overcomes it. The method maps the data matrices of different views into a shared space and then utilizes Gaussian mixture models to estimate the distributions of each cluster. By storing the parameters of the Gaussian mixture models, the class information of previous tasks is maintained. Experiments demonstrate its superiority over the compared ones.

**Strengths:**

1. The studied problem is interesting.

2. This paper is well-written, well-organized, and easy to follow.

3. Experiments show the excellent performance of the method, and the ablation study demonstrates the effectiveness of the critical components

**Weaknesses:**

1. In INTRODUCTION, the authors propose two challenges. However, I think they can be summarized as one challenge.
2. In NOTATION AND PROBLEM DEFINITION, it is obtained that each task has the same sample number $N$. It is more rational to set the sample number as $N_t$.
3. The method randomly selects samples to add to the buffer. The clustering performance might be sensitive to the selected samples. Also, since it is a random sampling strategy, the sample number in each cluster is not balanced. In extreme circumstances, there might be no samples in a cluster. I wonder if it affects the performance of the method.
4. The Gaussian mixture models and parameters share among views. MVC aims to extract consistent and complementary information among views. The assumption that the data of different views have a similar distribution might be unsuitable.
5. Some recent works could be discussed, such as [1].
6. The used datasets are small, and the time efficiency could be compared. I wonder if the method can address large-scale data.
[1] Li, D., Wang, T., Chen, J., Kawaguchi, K., Lian, C., & Zeng, Z. (2023). Multi-View Class Incremental Learning. ArXiv, abs/2306.09675.

**Questions:**

1. In INTRODUCTION, the authors propose two challenges. However, I think they can be summarized as one challenge.
2. In NOTATION AND PROBLEM DEFINITION, it is obtained that each task has the same sample number $N$. It is more rational to set the sample number as $N_t$.
3. The method randomly selects samples to add to the buffer. The clustering performance might be sensitive to the selected samples. Also, since it is a random sampling strategy, the sample number in each cluster is not balanced. In extreme circumstances, there might be no samples in a cluster. I wonder if it affects the performance of the method.
4. The Gaussian mixture models and parameters share among views. MVC aims to extract consistent and complementary information among views. The assumption that the data of different views have a similar distribution might be unsuitable.
5. Some recent works could be discussed, such as [1].
6. The used datasets are small, and the time efficiency could be compared. I wonder if the method can address large-scale data.
[1] Li, D., Wang, T., Chen, J., Kawaguchi, K., Lian, C., & Zeng, Z. (2023). Multi-View Class Incremental Learning. ArXiv, abs/2306.09675.

---

### Meta-Review · Area_Chair_K7Zx · 2023-12-11

**Metareview:**

This paper received consistent negative ratings, and the authors didn't provide any feedback during the rebuttal period. Therefore, this is a clear rejection.

**Justification For Why Not Higher Score:**

This paper received consistent negative ratings, and the authors didn't provide any feedback during the rebuttal period.

**Justification For Why Not Lower Score:**

N/A

---

### Decision · Program_Chairs · 2024-01-16

Reject